# Acceptance and hesitancy of COVID-19 vaccine among Nepalese population: A cross-sectional study

**Suresh Dahal**[1]*, **Srishti Pokhrel**[1], **Subash Mehta**[1], **Supriya Karki**[1], **Harish Chandra Bist**[1], **Dikesh Kumar Sahu**[1], **Nimesh Lageju**[2], **Sagar Panthi**[2], **Durga Neupane**[2], **Ashish Shrestha**[1], **Tarakant Bhagat**[1], **Santosh Kumari Agrawal**[1], **Ujwal Gautam**[1]

1 Department of Public Health Dentistry, B.P. Koirala Institue of Health Sciences, Dharan, Nepal,
2 Department of Internal Medicine, B.P. Koirala Institue of Health Sciences, Dharan, Nepal

* stddahal2@gmail.com

## Abstract

### Introduction

COVID-19 is an emerging infectious disease with a high transmission rate and substantial deaths. Various vaccines have been developed to combat it. This study is aimed to assess COVID-19 vaccine acceptance and hesitancy among the Nepalese population through a web-based survey.

### Materials and methods

This is a web-based cross-sectional descriptive study of Nepalese people 18 years and above from different regions of Nepal who use social media (Facebook, Twitter, Reddit) as well as instant messaging applications (Messenger, Viber, WhatsApp). The duration of the study was 3 months from 1st June 2021 to 31st August 2021. The sampling technique used was self-selected non-probability sampling. A validated questionnaire had been taken to record the data.

### Results

A total of 307 participants were included in the study. About three-fourths of participants 231 (75.2%) had not been vaccinated while 76 (24.8%) had been vaccinated with COVID -19 vaccine. Out of 231 non-vaccinated participants, most of participants 213 (92.2%) had shown acceptance of the COVID-19 vaccine. More than two-thirds of participants believed that the vaccine would protect them, their family members, and the community from having COVID-19 in the future. Very few participants 18 (7.2%) were hesitant to receive the vaccine against COVID-19. About two-thirds of participants were being afraid of adverse effects of the COVID-19 vaccine while more than half of participants hesitated due to lack of enough information regarding COVID-19 vaccine.

**Funding:** The authors received no specific funding for this work.

**Competing interests:** The authors have declared that no competing interests exist.

## Conclusion

This study can aid in the planning of vaccination campaigns and the direction of future public health efforts aimed at increasing COVID-19 vaccine uptake.

## Introduction

COVID-19 is an infectious illness that was discovered in December 2019 in Wuhan, Hubei Province, Central China. A quick outbreak with a high rate of transmission and significant mortality has already been documented globally, impacting 216 nations, regions, or territories [1, 2]. On March 11, 2020, the World Health Organization (WHO) proclaimed the COVID-19 outbreak a worldwide pandemic [3]. The first incidence was verified in Nepal on January 23, 2020. As of June 11, 2021, there have been 601,678 recorded cases and 8238 documented fatalities [4]. Several modeling studies were initiated during the early stages of the outbreak to assess the pandemic and the effectiveness of multiple population-wide strategies, such as lockdown, social distancing, quarantine, testing, contact tracing, and media-related awareness, among others, to mitigate COVID-19 spread [5, 6]. Nonetheless, despite such attempts, the epidemic remains uncontrollable. Although personal preventive measures taken by ordinary persons are critical to controlling the development of this infectious illness, immunization is a crucial protective strategy against COVID-19 [7].

Vaccines not only give individual protection to people who have been immunized, but they can also provide communal protection by limiting disease spread throughout a population. A SARS-CoV-2 vaccination that is successful will minimize morbidity and death while also allowing for significant easing of physical separation rules [8].

As of February 2021, over 70 vaccines had been tested in human clinical trials, including 20 in phase III studies [9, 10]. Furthermore, various vaccines have already been licensed in several countries, and immunization campaigns are underway in practically every region of the world. Understanding the public's worries about the COVID-19 vaccination is critical for planning optimal COVID-19 vaccine acceptance among the general public [11, 12].

Nepal's government has begun vaccination of front-liners with 1 million doses of COVID-19 AstraZeneca vaccines AZD1222 (Covishield®) supplied by India's government under the Vaccine Maitri program, among other COVID-19 vaccinations authorized for broad distribution [13]. Similarly, immunization of Vero cells made by Sinovac Company China began on April 9, 2021, after China of Nepal contributed 1 million vaccinations. On June 28, 2021, the second dosage of Vero cell vaccination began [14].

The immunization against COVID-19 was supposed to be done in two stages. On January 27, 2021, the first phase of vaccination for frontline medical staff, sanitation workers, ambulance drivers, and security officials began in all 77 districts of Nepal. The first phase of the COVID-19 immunization program was completed on February 6, 2021, with about 184,857 persons receiving doses [15]. The effectiveness of this vaccination against symptomatic COVID-19 infection is reported to be 63.09% [16].

In one of the narrative reviews, data on COVID-19 vaccine acceptance rates were retrieved from surveys in 114 countries/territories where the acceptance rate of Nepal was among the highest (97%) in Asia along with Vietnam [17]. However, in a survey done among health care workers in Nepal, over one-third of the sample population was skeptical of the COVID-19 vaccine, indicating that research on these themes is critical for effective immunization campaigns [18].

A maximum of 2,861,314 vaccine shots has been delivered as of June 7, 2021 [19]. Vaccine acceptance encompasses a range of actions and views, ranging from outright denial of all vaccines to active support for vaccination recommendations. Vaccination hesitation is a subset of this spectrum in which people dispute the safety or need of a specific vaccine [20]. The World Health Organization (WHO) Strategic Advisory Group of Experts on Immunization (SAGE) defines vaccine hesitancy (VH) as a "delay in acceptance or refusal of immunization despite the availability of vaccination services" [11]. While ensuring the effective and fair delivery of vaccinations is a significant governmental concern, ensuring population acceptability is as critical. Acceptance of vaccinations and faith in the organizations that deliver them are likely to be major factors of any immunization campaign's success [21].

Estimation of vaccine acceptance rates can be helpful to plan actions and intervention measures necessary to increase awareness and assure people about the safety and benefits of vaccines, which in turn would help to control virus spread and alleviate the negative effects of this unprecedented pandemic [22, 23].

Several studies have been conducted to assess the public views on the COVID-19 vaccination, as well as vaccine hesitancy. Different national surveys, as well as global surveys, have been done to assess the acceptance and hesitancy of the COVID-19 vaccine [24]. Evaluation of attitudes and acceptance rates towards COVID-19 vaccines can help to initiate communication campaigns that are much needed to make the vaccination program successful [25].

The objective of this study was to assess COVID-19 vaccine acceptance and hesitancy among the Nepalese population through a web-based survey. The results of this study can be utilized in planning vaccination campaigns and guiding future public health efforts that aim to increase the uptake of COVID-19 vaccines.

## Materials and methods

### Study setting, population and design

This is a web-based cross-sectional descriptive study comprised of Nepalese people from different regions of Nepal who use social media (Facebook, Twitter, Reddit) and instant messaging applications (WhatsApp, Viber). The sampling technique used was self-selected non-probability sampling. Nepalese people from age 18 years and above who were not vaccinated, and who were not willing to take were included. People who didn't use social media, who were under 18 years of age, and who didn't give consent for the study were excluded. The duration of the study was 3 months from 1st June 2021 to 31st August 2021.

### Study tools

A variety of tools, guidelines and other material were examined and reviewed for the formulation of questionnaires. Following that, the authors collaborated to create a draft questionnaire, which was then evaluated by study team members, topic specialists, researchers, and policymakers to guarantee the content validity. Priority was given to information considered useful to the general public when constructing the surveys. We chose realistic and commonly encountered questions for the surveys based on the experiences of experts working in COVID-19 Hospitals. The questionnaire was then pre-tested among 10 random people, and required adjustments and amendments were made in the final form, such as simplifying the wording and adding the Nepali translation of the questionnaire. Two people who were proficient in both languages translated the English questionnaire into Nepali. Back-translation of the Nepali version of the questionnaire into English was used to guarantee that the original meaning of the questions was retained. Subject specialists examined the questionnaire's facade and consensual validity. The questionnaire was made available in both languages.

The questionnaire consisted of 4 sections based on demographic profile, vaccination status, willingness for vaccination, and hesitancy towards it respectively. The first section included information about gender, age, section, education level, and occupation of each respondent to examine heterogeneity across demographic strata. The second section included vaccinated status, and whether the respondent had been vaccinated or unvaccinated. The third section and fourth section consisted of six and seven Likert scale-based questions asking the reasons for acceptance or hesitancy of the COVID-19 vaccine respectively.

Then the final questionnaire was distributed to the people randomly via social media (Facebook, Twitter, Reddit) and instant messaging applications (WhatsApp, Viber) in the form of Google sheets.

## Outcome measures

The outcome measures were the willingness of the Nepalese population for vaccination against COVID-19, reasons for acceptance of COVID-19 vaccine, and reasons for the hesitancy of COVID-19 vaccine.

## Sample size calculation

In this study, p = 13.7% (Based on estimate of prevalence of hesitancy towards COVID-19 vaccine among general population in India according to Khan et al. [27]) z = 1.96 (at 95% confidence level), d = 5% (Absolute error). where, n = Sample Size; p = Proportion of the event in the population; q = 1-p, d = Acceptable margin of error in estimating the true population proportion; Z = value at 95% confidence level.

Now using the following formula to estimate the sample size, $= Z^{2*}p^*q/d^2 = 181.6 \sim 181$

Adding 10% for non-respondents: Sample size = 181 + 10% of 181 = 199.1 ~ 200

The total sample size as calculated was 200. However, we collected as many responses as we could.

## Ethics statement

Informed consent was obtained from all the participants before enrolling in the study. The information collected from this research project was kept confidential. At no point in time, any of the information was disclosed outside the investigator's circle.

The study was approved by the Institute Review Committee, B. P. Koirala Institute of Health Sciences, Dharan. All study was carried out in line with relevant guidelines and regulations.

## Data analysis

The data collected was imported from Google sheets and then transferred into SPSS (Statistical Package for Social Sciences) v.11.5 software for statistical analysis. Graphical methods like bar diagrams and tabular presentations were used to describe different categorical variables. Frequency and proportion were calculated to describe the categorical variables of the study. Mean (standard deviation) was calculated to describe the continuous variables of the study. For skewed data, median (IQR) was presented.

# Results

Among 325 respondents in the online study, 320 (98.2%) people agreed to participate in it.

Out of 320 participants, 307 were only included for analysis while the remaining were filtered as per exclusion criteria (11 were of age<18 years, 2 were of Indian nationality). More

than half of the respondents 173 (56.4%) were male. The mean age of the participants was 24.15 (6.8). Most of the participants were from province 1 (21.8%). Based on the geographical distribution, almost half of the participants 54.7% were from the Terai region and had a graduation level of education (50.2%). In terms of occupation, the majority of them were students 190 (61.9%). (Table 1)

Vaccination status:

Out of 307 participants, about three fourth of participants 231 (75.2%) had not been vaccinated while 76 (24.8%) had been vaccinated with COVID -19 vaccine.

Out of 231 non-vaccinated participants, most of the participants 213 (92.2%) had shown acceptance towards COVID-19 vaccine if made available however very few have shown 18 (7.8%) hesitancy to get it.

Among the acceptance group (213), participants agreed to several factors for acceptance. More than two-thirds of the participants 179 (84%) completely agreed that the vaccine would protect them from having COVID-19 in the future and nearly half of the participants 104 (48.8%) completely agreed that the vaccine would decrease the complications from COVID-19. Similarly, more than three fourth of the participants 172 (80.8%) agreed that vaccination would protect their family members and also agreed that vaccination would protect the community against having COVID-19 in the future165 (77.5%).

More than half of the participants 111 (52.1%) completely agreed that getting vaccinated would ease the precautionary measure including lockdown, quarantine, and travel ban, and would make daily life normal while very few 3.8% disagreed.

The results are shown in Fig 1.

**Table 1. Sociodemographic data of participants(n = 307).**

| Variables | | N (%) |
|---|---|---|
| **Socio-demographic profile** | | |
| 1. Gender | Male | 173 (56.4%) |
| | Female | 134 (43.6%) |
| 2. Mean age (SD) | Acceptance mean age | 24.37 (6.9) |
| | Hesitant mean age | 21.56 (2.8) |
| | Total mean age | 24.15 (6.8) |
| 3. Province distribution | Province 1 | 67 (21.8%) |
| | Madhesh Province | 26 (8.5%) |
| | Bagmati Province | 59 (19.2%) |
| | Gandaki Province | 48 (15.6%) |
| | Lumbini Province | 54 (17.6%) |
| | Karnali Province | 3 (1%) |
| | Sudur Paschim Province | 50 (16.3%) |
| 4. Geographical distribution | Mountain region | 22 (7.2%) |
| | Hilly region | 117 (38.1%) |
| | Terai region | 168 (54.7%) |
| 5. Education level | Informal education | 5 (1.6%) |
| | Secondary | 10 (3.3%) |
| | Higher secondary | 107 (34.9%) |
| | Graduate | 154 (50.2%) |
| | Above graduate | 31 (10.1%) |
| 6. Occupation | Employed | 104 (33.9%) |
| | Unemployed | 13 (4.2%) |
| | Students | 190 (61.9%) |

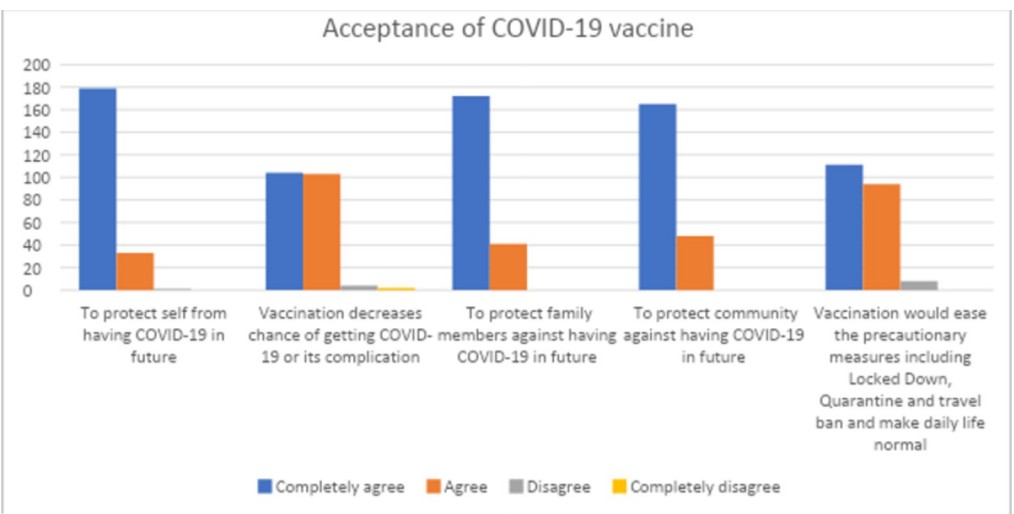

**Fig 1. Acceptance of COVID-19 vaccine among Nepalese population.**

The hesitancy of COVID-19 vaccine:

Very few participants 18 (7.2%) were hesitant to receive the vaccine against COVID-19.

More than half 11 (61.1%) agreed that they were afraid of the queue in the vaccination center while 5 participants (27.8%) disagreed with it. Similarly, about two-thirds of participants 12 (66.7%) agreed on being afraid of adverse effects of the COVID-19 vaccine while 2 participants (11.1%) disagreed on it and more than half of participants 11 (61.1%) agreed on not having enough information about COVID-19 vaccine while very few participants 2 (11.1%) disagreed on it. On the contrary, more than half of participants 10 (55.6%) disagreed on the COVID-19 vaccine not being considered serious and no need to take vaccine while less than one-fourth of participants 3 (16.7%) agreed on it. Similarly, almost half of the participants 8 (44.4%) disagreed on believing that natural immunity is sufficient so they don't think that they need vaccine while about one-third of the participants 7 (38.9%) agreed on it and more than two-thirds of the participants 13 (72.2%) disagreed on not having time to get vaccinated while 4 participants (22.2%) agreed on it.

The results are shown in Fig 2.

## Discussion

This is a cross-sectional study conducted to assess the acceptance and hesitancy for the COVID-19 vaccine among the general population of Nepal and to our knowledge it is the first study of this kind conducted in the general population of Nepal. A total of 325 people participated in this study.

Although numerous modeling studies were undertaken during the early stages of the COVID-19 pandemic to limit the transmission, vaccination was proven to be the most effective method to minimize the spread as well as lessen illness consequences [9]. Vaccinations are largely regarded as one of the most efficient public-health preventative strategies [26]. However, vaccination reluctance among not just individuals but even medical professionals have been an issue in recent years. Vaccine reluctance varies by time, location, and vaccine type, and is impacted by a number of factors. As a result, it is required to analyze vaccination acceptability of the COVID-19 vaccine and the variables influencing it in each location in order to organize educational initiatives to promote vaccine acceptance [11]. Through this study, we have tried to understand various factors of acceptance or hesitancy for the COVID-19 vaccine.

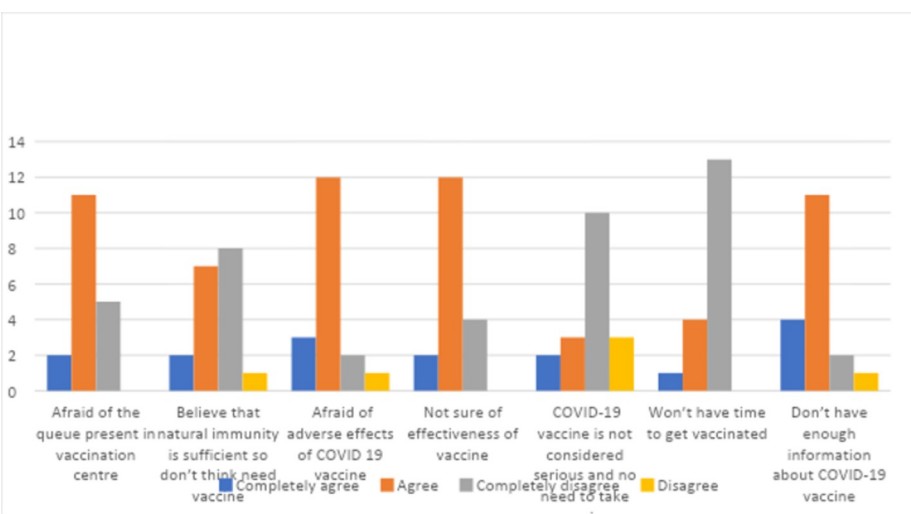

**Fig 2. Hesitancy for COVID-19 vaccine.**

The current study showed that almost all the participants were willing to get vaccinated if made available. These observations are similar to findings of a study conducted among the general population of India [27] using an online self-administered questionnaire where the acceptance rate was 86.3% and a similar result was observed in lower-middle-income countries [24].

Similarly, 71.5% of participants in a global study of 19 nations said they would accept vaccination if it was demonstrated to be effective and safe [28]. This proportion is lower than what we discovered in our investigation. Although it is difficult to compare our study to earlier ones due to variations in how the questions were given and the timeframe of the questionnaires.

Few trials, however, conducted in Kuwait [29], Jordan [30], revealed the lowest percentages of acceptability of the COVID19 vaccination, 51.3%, and 37.45%, respectively. This disparity might be attributed to a decrease in the vaccine's putative efficiency but acceptance would have been greatest for a hypothetical vaccination with a 95% efficacy rate. However, another explanation for such low rates of COVID-19 acceptance in these Middle Eastern countries was linked to high embrace of conspiracy beliefs regarding COVID-19 vaccination [31].

The current study showed that the participants who were acceptable to the vaccine, more than two-thirds of the participants completely agreed that the vaccine would protect them from having COVID-19 in the future. Likewise, a handful of studies showed participants wanted to get vaccinated to protect themselves from the COVID-19 [24]. Almost half of the population who accepted the vaccine have completely agreed that getting COVID -19 decreases the complications of COVID-19 consistent with other studies [24]. Similar to this, a study done in Kuwait shows the self-perceived severity of symptoms, participants who anticipated that their COVID-19 symptoms would be mild were less accepting of the vaccine than those who anticipated that their symptoms would be severe (55.8 vs. 63.9%) [29].

In the present study, participants who were willing to get vaccinated completely agreed that it would protect family members and the community. Similar to this result, willingness to protect others by getting oneself vaccinated, was also reported to be one of the important factors associated with COVID-19 vaccine acceptance in a previous study [32].

The majority of participants accepting vaccination agreed that vaccination would ease the precautionary measures including locked down, quarantine, and travel ban and make daily life normal, which was similar to the result of the study where acceptors were more likely to

mention reducing the risk of COVID-19 infection and benefits related to livelihoods, and re-starting economic activities and getting back to normal life [32].

The assumed prevalence, used for sample size calculation, of vaccine hesitancy towards the COVID-19 vaccine among the general population in India according to Khan et al. [27] was 13.7%. Upon collecting the data and analysis, our study also showed a low prevalence (7.2%) of vaccine hesitancy.

Our study observed hesitancy for the COVID-19 vaccination, more than two-thirds of the participants who hesitated were afraid of unknown adverse effects of the COVID-19 vaccine which is similar to other studies done among health care workers in Bangladesh (87.3%) [33], general population of African American population (64.1%) [34], the general population in Malta (85.1%) [35], general populations in India (64.4%) [27] and population of Israel (70%) [36].

Moreover, almost half of the participants agreed that natural immunity is sufficient against COVID-19 which was the reason for their hesitancy. Similarly, a study done in Jordan showed participants to achieve immunity against COVID-19 using natural way was the most commonly reported reason to refuse vaccination (64%) [37]. In contrast, a study showed less than 10% considering natural immunity and traditional remedies rather than vaccination [35]. Also, a study showed only 29.1% of similar results [34].

More than two-thirds of participants disagreed to consider COVID-19 not being a serious disease which showed similarity to the study where less than 10% population considered COVID-19 as flu which is not serious [35].

The majority of the hesitant population were not sure of the effectiveness of COVID-19 vaccination consistent with the study where more than half (68%) did not trust the vaccine to be effective and 60% did not trust the pharmaceutical companies [27]. Also a study done in Kuwait showed doubtful efficacy as a reason for hesitancy (69.9%) [29]. In contradiction, a study done in Israel showed 20% of the population is concerned about the effectiveness of vaccines [35]. The probable reason for hesitancy could be the lack of enough information about the COVID-19 vaccine [29, 30].

Collectively, these results highlight the need for improving public knowledge and trust in the effectiveness of vaccines against infectious diseases and their safety.

## Limitations

Because our study sample was limited, the generalizability of our findings may be impeded by the uncertain representativeness of our study sample due to the nonrandom sampling procedure utilized. Another drawback of our study is that participants had to have access to a smart-phone, tablet, or computer in order to participate, which might have created a selection bias. Furthermore, we wanted to examine people's willingness to receive a vaccine at a time when just the first phase of immunization had been completed. As a result, as additional evidence about the safety and efficacy of COVID-19 vaccines becomes available, people's attitudes toward vaccination may shift. Our study did not take into consideration psychological elements and their effect on vaccination adoption, such as faith in science, which was found to have decreased during the COVID-19 epidemic by an Italian study [38]. Nonetheless, our study investigated a wide variety of parameters regarding the acceptability of COVID-19 immunization, which may aid in guiding future public health activities aimed at increasing COVID-19 vaccine uptake.

## Conclusion

This study assessed the acceptance and hesitancy for the COVID-19 vaccine among the general population of Nepal via an online form. The majority of the participants accepted the vaccine

(92.2%). The factors for the acceptance were mainly to protect themselves, their family, and their community from COVID-19. Also, most of the participants agreed that vaccination could decrease the complications of COVID-19. The majority of participants (96.2%) agreed that vaccination would ease the locked down and travel ban.

Also, few of the participants (7.2%) hesitated for vaccination due to factors which are unknown adverse effects of the COVID-19 vaccine, and doubtful effectiveness of the vaccine. However almost half of the participants disagreed with their hesitancy because natural immunity is considered sufficient against COVID-19.

The major findings of this study can be utilized in planning vaccination campaigns furthermore the level of vaccine acceptance can be increased within the population if additional studies can confirm to the safety and effectiveness of the available vaccine candidates.

## Supporting information

**S1 File. Questionnaire.**
(DOCX)

## Author Contributions

**Conceptualization:** Suresh Dahal, Srishti Pokhrel, Subash Mehta, Supriya Karki, Harish Chandra Bist, Dikesh Kumar Sahu, Sagar Panthi, Ashish Shrestha, Tarakant Bhagat, Santosh Kumari Agrawal, Ujwal Gautam.

**Data curation:** Suresh Dahal, Srishti Pokhrel, Subash Mehta, Supriya Karki, Harish Chandra Bist, Dikesh Kumar Sahu, Sagar Panthi, Ashish Shrestha, Tarakant Bhagat, Santosh Kumari Agrawal, Ujwal Gautam.

**Formal analysis:** Suresh Dahal, Srishti Pokhrel, Supriya Karki, Nimesh Lageju, Durga Neupane.

**Investigation:** Suresh Dahal, Srishti Pokhrel, Supriya Karki, Harish Chandra Bist, Dikesh Kumar Sahu, Sagar Panthi.

**Methodology:** Suresh Dahal, Srishti Pokhrel, Subash Mehta, Supriya Karki, Harish Chandra Bist, Dikesh Kumar Sahu, Nimesh Lageju, Sagar Panthi, Durga Neupane, Ashish Shrestha, Tarakant Bhagat, Santosh Kumari Agrawal, Ujwal Gautam.

**Project administration:** Srishti Pokhrel, Subash Mehta, Supriya Karki, Harish Chandra Bist, Dikesh Kumar Sahu, Sagar Panthi, Ashish Shrestha, Tarakant Bhagat, Santosh Kumari Agrawal, Ujwal Gautam.

**Software:** Nimesh Lageju, Durga Neupane.

**Supervision:** Suresh Dahal, Srishti Pokhrel, Ashish Shrestha, Tarakant Bhagat, Santosh Kumari Agrawal, Ujwal Gautam.

**Validation:** Suresh Dahal, Srishti Pokhrel, Subash Mehta, Supriya Karki, Harish Chandra Bist, Dikesh Kumar Sahu, Nimesh Lageju, Sagar Panthi, Durga Neupane, Ashish Shrestha, Tarakant Bhagat, Santosh Kumari Agrawal, Ujwal Gautam.

**Visualization:** Suresh Dahal, Srishti Pokhrel, Subash Mehta, Supriya Karki, Harish Chandra Bist, Dikesh Kumar Sahu, Ashish Shrestha, Tarakant Bhagat, Santosh Kumari Agrawal, Ujwal Gautam.

**Writing – original draft:** Suresh Dahal, Supriya Karki.

**Writing – review & editing:** Nimesh Lageju, Sagar Panthi, Durga Neupane.

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
