## [Decision Letter · Decision Letter 0]

4 Sep 2022

PONE-D-22-19860Acceptance and hesitancy of COVID-19 vaccine among Nepalese population: a cross-sectional studyPLOS ONE

Dear Authors, 

Thank you for submitting your manuscript to PLOS ONE. After careful consideration, we feel that it has merit but does not fully meet PLOS ONE’s publication criteria as it currently stands. Therefore, we invite you to submit a revised version of the manuscript that addresses the points raised during the review process.

We look forward to receiving your revised manuscript.

Kind regards,

Yaser Mohammed Al-Worafi

Academic Editor

PLOS ONE

Journal Requirements:

a) Did participants provide their written or verbal informed consent to participate in this study?

Reviewers' comments:

Reviewer's Responses to Questions

**Comments to the Author**

1. Is the manuscript technically sound, and do the data support the conclusions?

Reviewer #1: Yes

Reviewer #2: Partly

2. Has the statistical analysis been performed appropriately and rigorously? 

Reviewer #1: Yes

Reviewer #2: No

3. Have the authors made all data underlying the findings in their manuscript fully available?

Reviewer #1: Yes

Reviewer #2: No

4. Is the manuscript presented in an intelligible fashion and written in standard English?

Reviewer #1: Yes

Reviewer #2: Yes

5. Review Comments to the Author

Reviewer #1: Thanks for the invitation to review this manuscript.

In the current study, Suresh Dahal et al. investigated COVID-19 vaccine hesitancy among adult Nepalese population using a cross-sectional web-based survey. The study results pointed to a low rate of COVID-19 vaccine hesitancy in the country compared to rates observed in other regions and countries worldwide.

Overall, the manuscript is well-prepared, and the study design was appropriate to reach reliable conclusions regarding the study objectives.

I have the following minor comments that hopefully can help the authors to improve the final manuscript:

Abstract: The authors can benefit from adding the cited reasons reported by the participants who were hesitant to COVID-19 vaccination.

Introduction: It was clear and provided sufficient background on the study topic and included all relevant references.

Methods: The study design was appropriate, and the methods were described in enough details to allow replication of the study. One important point that should be clarified by the authors is the exact wording of the survey item that assessed COVID-19 vaccine acceptance/hesitancy besides the possible answers. I could not access the supplementary file; therefore, I am not sure if the authors provided the complete questionnaire. If not, the authors are encouraged to do so.

Results: The study results were presented clearly including the table and figures.

Discussion and Conclusions: The study results were presented in the context of the extant literature properly and the conclusions were supported by the results. Importantly, the authors presented the potential limitations of the study.

Thanks!

Reviewer #2: This is a prevalence study, looking at vaccine hesitancy.

A few comments worth addressing.

Can the authors elaborate on the why the final questionnaire was assigned randomly to people - does this mean some did not get to complete this questionnaire? What was the random method used.

Can the authors expand on the outcome measures, was this calculated by a total /mean score from the six and seven Likert questionnaire?

Sample size was set at 200, the authors noted that, they however, collected as many responses, was this still done in the recruitment period? As it might be misleading to over collect data, more than is needed for the study.

Under the data analysis section, you may want to add, for skewed data, median (IQR) were presented.

Under data analysis, remove 95%CI, as this was not done/represented, either add 95% results or omit.

Line 210- The mean “(SD)” age of the participants was 24.15 (“6.8”).- changed to 1 d.p - be consistent throughout manuscript.

Table 1 - format so that SD is 1 d.p.

Worth mentioning in the discussion that the assumed prevalence of vaccine hesitancy used for the sample size calculator, upon collecting data in the chosen population in this study, low prevalence of vaccine hesitancy was observed.

6. PLOS authors have the option to publish the peer review history of their article (what does this mean?). If published, this will include your full peer review and any attached files.

Reviewer #1: No

Reviewer #2: No

---

## [Author Response · Author response to Decision Letter 0]

10 Sep 2022

Review comments to the authors:

Reviewer #1: 

Reviewer Comments: 

Thanks for the invitation to review this manuscript.

In the current study, Suresh Dahal et al. investigated COVID-19 vaccine hesitancy among adult Nepalese population using a cross-sectional web-based survey. The study results pointed to a low rate of COVID-19 vaccine hesitancy in the country compared to rates observed in other regions and countries worldwide.

Overall, the manuscript is well-prepared, and the study design was appropriate to reach reliable conclusions regarding the study objectives.

I have the following minor comments that hopefully can help the authors to improve the final manuscript:

Abstract: The authors can benefit from adding the cited reasons reported by the participants who were hesitant to COVID-19 vaccination.

Introduction: It was clear and provided sufficient background on the study topic and included all relevant references.

Methods: The study design was appropriate, and the methods were described in enough details to allow replication of the study. One important point that should be clarified by the authors is the exact wording of the survey item that assessed COVID-19 vaccine acceptance/hesitancy besides the possible answers. I could not access the supplementary file; therefore, I am not sure if the authors provided the complete questionnaire. If not, the authors are encouraged to do so.

Results: The study results were presented clearly including the table and figures.

Discussion and Conclusions: The study results were presented in the context of the extant literature properly and the conclusions were supported by the results. Importantly, the authors presented the potential limitations of the study.

Response from Author: Thank you very much for reviewing the manuscript and providing constructive feedback. The minor comment on the abstract has been addressed. (Page number 3, line number: 60-62)

A complete questionnaire (both in English and Nepali language) is submitted as a supplementary file. We hope the reviewer’s comment regarding the exact wording of the survey item that assessed COVID-19 vaccine acceptance/hesitancy besides the possible answers would be addressed via the submitted questionnaire (as a supplementary file).

Reviewer #2:

Reviewer Comments: Can the authors elaborate on the why the final questionnaire was assigned randomly to people - does this mean some did not get to complete this questionnaire? What was the random method used.

Response from Author: The questionnaire was assigned randomly to people in contact with authors via social media and instant messaging apps in various parts of the country so that responses from people of diverse ages, professions, educational qualifications, and regions could be included as a part of the study. All of the shared questionnaires were not responded to, but those who had responded were complete. However, among the respondents, very few were excluded based on exclusion criteria. (Page number 10, Line number: 213-214) 

The sampling technique used was self-selected non-probability sampling.

Reviewer Comments: Can the authors expand on the outcome measures, was this calculated by a total /mean score from the six and seven Likert questionnaires?

Response from Author: The outcome measures of the study show the percentage of acceptance among the Nepalese population for vaccination against COVID-19. It also depicts the different reasons for acceptance. Similarly, it shows the percentage of hesitancy among the Nepalese population for vaccination against COVID-19 along with the different reasons for their hesitancy. 

The questionnaire regarding acceptance had 6 questions and regarding hesitancy had 7 questions. All the questions were measured by a 4-unit Likert Scale including, completely agree, Agree, Disagree, and Completely Disagree. The table shows the sum of responses on each option and the percentage was calculated accordingly. 

Reviewer Comments: Sample size was set at 200, the authors noted that they, however, collected as many responses, was this still done in the recruitment period? As it might be misleading to over collect data, more than is needed for the study.

Response from Author: The sample size was calculated based on the estimate of the prevalence of hesitancy toward COVID-19 vaccine among the general population in India according to Khan et al. However, we had supplied the online questionnaire to more people for assurance if some of them might be reluctant to participate and fill up the questionnaire. Also, the more response or size of the data could help us collect more diverse data and strengthen our results and conclusion. All these data were collected only during the recruitment period. After the recruitment period was completed, the online questionnaire was made unavailable.

Reviewer Comments: Under the data analysis section, you may want to add, for skewed data, median (IQR) were presented.

Under data analysis, remove 95%CI, as this was not done/represented, either add 95% results or omit.

Response from Author: Under the data analysis section, the following statement has been added. “For skewed data, median (IQR) was presented.” (Page number 9, Line number 208)

Under the data analysis section, 95% CI has been omitted.

Reviewer Comments: Line 210- The mean “(SD)” age of the participants was 24.15 (“6.8”).- changed to 1 d.p – be consistent throughout manuscript.

Table 1 - format so that SD is 1 d.p.

Response from Author: The mean “SD” age of the participants at (Page number 10, Line number: 215) along with other values in Table 1 has been corrected and made 1 d.p. so that the data are consistent throughout the manuscript. 

Reviewer Comments: Worth mentioning in the discussion that the assumed prevalence of vaccine hesitancy used for the sample size calculator, upon collecting data in the chosen population in this study, low prevalence of vaccine hesitancy was observed.

Response from Author: The above author’s comment has been addressed in the discussion section. (Page number 14, Line number: 301-304)

---

## [Editor Report · Decision Letter 1]

26 Sep 2022

Acceptance and hesitancy of COVID-19 vaccine among Nepalese population: a cross-sectional study

PONE-D-22-19860R1

Dear Authors, 

We’re pleased to inform you that your manuscript has been judged scientifically suitable for publication and will be formally accepted for publication once it meets all outstanding technical requirements.

Kind regards,

Yaser Mohammed Al-Worafi

Academic Editor

PLOS ONE

---

## [Editor Report · Acceptance letter]

28 Sep 2022

PONE-D-22-19860R1 

Acceptance and hesitancy of COVID-19 vaccine among Nepalese population: a cross-sectional study 

Dear Dr. Dahal:

I'm pleased to inform you that your manuscript has been deemed suitable for publication in PLOS ONE. Congratulations! Your manuscript is now with our production department. 

Kind regards, 

on behalf of

Professor Yaser Mohammed Al-Worafi 

Academic Editor

PLOS ONE